# Evaluation of the Effects of Food Safety Training on the Microbiological Load Present in Equipment, Surfaces, Utensils, and Food Manipulator’s Hands in Restaurants

**DOI:** 10.3390/microorganisms12040825

**Published:** 2024-04-19

**Authors:** Miguel Castro, Kamila Soares, Carlos Ribeiro, Alexandra Esteves

**Affiliations:** 1Department of Veterinary Science, School of Agrarian and Veterinary Science (ECAV), University of Trás-os-Montes e Alto Douro (UTAD), 5000-801 Vila Real, Portugal; kamilacgsoares@gmail.com; 2Veterinary and Animal Research Center (CECAV), University of Trás-os-Montes e Alto Douro (UTAD), 5000-801 Vila Real, Portugal; 3Department of Agronomy, School of Agrarian and Veterinary Science (ECAV), University of Trás-os-Montes e Alto Douro (UTAD), 5000-801 Vila Real, Portugal; cribeiro@utad.pt; 4Centre for the Research and Technology of Agro-Environmental and Biological Sciences, Centre for the Research and Technology of Agro-Environmental and Biological Sciences (CITAB), 5000-801 Vila Real, Portugal; 5Inov4Agro Associated Laboratory, 5000-801 Vila Real, Portugal; 6Al4AnimalS Associated Laboratory for Animal and Veterinary Science, 5000-801 Vila Real, Portugal

**Keywords:** coagulase-positive *Staphylococcus*, food safety training, food handlers, *Listeria monocytogenes*, microbiological analysis, restaurants

## Abstract

Training food handlers is essential to ensure food safety. However, the efficacy of training programs relying solely on theoretical information remains uncertain and often fails to induce significant changes in inadequate food practices. Training programs in good hygiene and food safety practices that integrate theoretical and practical approaches have emerged as a vital tool, enabling food handlers to apply their knowledge during work hours and clarify doubts. This study aimed to assess the impact of food safety training based on theoretical and on-the-job training on the microbiological counts of equipment, surfaces, utensils, and food handler (FH) hands. The hygiene and food safety conditions of four restaurants were analyzed through facility checklists, employee questionnaires, and microbiological analyses conducted before and after training. Eight sample collection moments were conducted at each restaurant before and after training. The pre-training results indicate that 15% and 26% of analyses for *Enterobacteriaceae* and total mesophilic aerobic bacteria (TMB), respectively, did not comply with hygiene safety limits. Additionally, 31% and 64% of *Enterobacteriaceae* and TMB values, respectively, exceeded safety limits on food handler hands. Positive cases of coagulase-positive *Staphylococcus* (CoPS) resulted from unprotected wounds on some FH hands. The presence of *Listeria monocytogenes* in drains was also identified as a concern. Following training, significant differences in results were observed. In many cases, there was a reduction of over 80% in microbial load for *Enterobacteriaceae* and TMB collected from equipment, surfaces, utensils, and food handler hands. The presence of *L. monocytogenes* in drains was also eliminated after food safety training. In conclusion, this study underscores the importance of effective training in improving food safety practices.

## 1. Introduction

Food safety and foodborne diseases (FBDs) are important, as they pose significant concerns for public health and result in millions of dollars in healthcare expenses each year [1,2]. FBDs often occur in restaurants due to improper handling, preparation, storage, transportation, and sanitation. They can frequently be caused by bacteria, viruses, parasites, chemical contaminants, and allergens [3]. Inadequate hygiene and sanitation standards increase the likelihood of foodborne infections. In 2022, 5763 foodborne outbreaks were registered in the European Union, which corresponds to a 44% increase when compared with 2021 [4]. According to an investigation by the Food and Nutrition Department of the National Institute of Health Ricardo Jorge (INSA), in Portugal in 2017, the year to which the most recent data refers, 18 outbreaks were reported, of which 8 were reported from school canteens and bars, cafes, restaurants, and hotels. In 13 of the 18 reported outbreaks, the identification of the pathogenic agent was possible since the toxin-producing bacteria (12/18) were the most prevalent. The main factors referred to in this study that contributed to the occurrence of outbreaks of foodborne illnesses were inadequate thermal treatment, abuse of time/temperature, cross-contamination, and the use of unsafe raw materials [5]. Hands of FHs have also been considered an important determinant of microbiological contamination [6].

Therefore, it is crucial to emphasize the verification, implementation, and monitoring codes of good manufacturing practices (GMP) [2,4].

The European Community has implemented food safety regulations and laws to ensure compliance with the hygiene of foodstuffs (Reg. No. 852/2004) [7]. In Portugal, the Law No. 10/2015 [8] mandates that food and beverage establishments must upkeep their facilities, equipment, furniture, and utensils in continuous good condition and hygiene; adhere to relevant legal and regulatory standards for handling, preparing, packaging, and selling food products; and grant competent supervisory authorities access to the establishment for inspecting related documents, books, and records.

It is very common to find a lack of implemented HACCP plan or a poorly implemented HACCP plan in small restaurant units, primarily due to insufficient funding, managers’ inadequate grasp of the legal documentation and adherence to prerequisites, and a general lack of knowledge regarding food safety [9,10]. In many instances, small-scale restaurants may prioritize survival in the fiercely competitive market over investing in food safety measures. Consequently, resources may be allocated to other areas, with food safety being relegated to a secondary concern.

Although FHs typically receive food safety training, it is often purely theoretical. As a result, some of them may exhibit poor practices while working, such as inadequate cleaning of surfaces, equipment, utensils, and hands; cross-contamination between raw and cooked foods; improper thawing of frozen food; and incorrect storage [11]. The FH must receive a combination of theoretical knowledge and practical training to enhance their ability to learn and improve their daily food safety practices [12,13].

Microorganisms indicating poor hygiene practices, such as *Enterobacteriaceae* (ENT), can be used to assess hygienic conditions. Bacteria belonging to the *Enterobacteriaceae* family are the main agents of intestinal infection, with *Escherichia coli* being the most relevant microorganism in food. Food-associated outbreaks have been particularly linked to verocytotoxin-producing *E*. *coli* (VTEC), with the O157:(H7) strain being recognized as a highly significant cause of foodborne illness. The presence of a large microbial population responsible for food contamination is evaluated through the count of mesophilic aerobic bacteria (TMB) (colonies that grow at 30 °C). Although there is no distinction between pathogenic microorganisms and those causing changes in food, determining mesophilic aerobic bacteria provides a general indication of food or utensil contamination during storage or processing. Therefore, mesophilic aerobic bacteria, *E*. *coli*, and coagulase-positive *Staphylococcus* have been considered microbial indicators and used to assess the hygienic and sanitary quality of food [6]. Other good indicators of hygienic and sanitary conditions are yeast and molds. Yeast (Y) reproduces by budding, while mold (M) forms chains of microscopic cells with hyphae; both are capable of thriving in environments with low pH, moisture, or temperature, and high levels of salt or sugar [14], posing a problem in dry foods, salted fish, bread, pickles, fruits, preserves, jams, and similar commodities. They can produce mycotoxins that can contaminate human food at various stages in the food chain. Mycotoxins are heat stable within the range of conventional cooking temperatures (80–121 °C) and can cause mycotoxicosis, which can lead to an acute or chronic disease [15].

The objectives of this study in restaurants were as follows: (1) to analyze the importance of food safety and hygiene measures being correctly implemented through the completion of a checklist to analyze the parameters (physical facilities and environment; food handlers; equipment and utensils; reception and storage; preparation, cooking, pantry, and serving; quality control); (2) to assess the level of knowledge of food handlers through the completion of questionnaires; and (3) to determine the microbiological loads of surfaces, equipment, utensils, food handlers’ hands, and drains to evaluate the overall microbiological state of the restaurants. (4) To assess the influence of theoretical and on-the-job practical training on reducing overall microbial contamination, and on changing attitudes toward the acquisition of new knowledge, was another important objective of this study.

## 2. Materials and Methods

### 2.1. Experimental Design

The experimental design aimed to ascertain the microbial load across four restaurants following food safety training, as depicted in Figure 1. This design encompassed two analysis points: before and after theoretical and practical training sessions. Pre-training sampling occurred between November 2022 and February 2023, while post-training sampling took place between March 2023 and June 2023. The experimental design’s visual representation is available in Figure 1.

### 2.2. Restaurant Characterization

This study was conducted in four restaurants located in northern Portugal. The layouts of most of the restaurants under study did not have delineated physical separations between distinct areas, such as the demarcation of preparation of raw products, pastry area, and plating because of the small kitchen area. When the study was conducted, all establishments had implemented a food safety system based on the HACCP system according to Portuguese legislation. However, the creation of the HACCP plan was carried out by an external company, whose monitoring was not as active. Compliance with the records may sometimes not be applied correctly, leading to discrepancies between what occurred and what should be recorded.

### 2.3. Checklist Application

The restaurants under investigation were subjected to a checklist (Appendix A) based on Vieiros et al. (2007) [16]. In Table 1, the six modules present in the checklist are represented, namely, (1) physical facilities and environment; (2) food handlers; (3) equipment and utensils; (4) reception and storage; (5) preparation, cooking, pantry, and serving; and (6) quality control. Based on the results, the restaurants were scored, enabling us to assess the hygienic-sanitary conditions, providing an overview of the overall state of the establishment and food-handling practices.

The score of each item was obtained through the subsequent classification of their respective sub-items (Table 1). Thus, the classification of sub-items was done as follows:Yes (Y) when the proper adherence to a specific aspect was observed.No (N) when a particular sub-item did not comply.Not applicable (NA) whenever a certain procedure or equipment did not apply to the establishment.

Therefore, to calculate the score of each checklist module, it was necessary to know the total number of yes (Y) and not applicable (NA) outcomes for each module. First, the number of yes (Y) outcomes was multiplied by the module weight (W), which was obtained through the following formula:WM=(Total Y×W)/(K−Total NA)
where
WM—score of each module.Y—total number of yes outcomes.W—module weight.K—module constant (number of evaluated topics in each module).NA—total of not applicable outcomes.

To transform the score of each module into a percentage and enable the classification of the modules and the checklist, the following formulas were used:%Module=(WM×100)/K
%Global=∑(WM×100)/100

These formulas were used to calculate the percentage of each module and the overall percentage, facilitating the qualitative classification (Table 2) of the restaurant units under study.

### 2.4. Structure of the Questionnaire about the Level of Knowledge

For this evaluation, each staff member completed a questionnaire to evaluate their knowledge of food safety protocols (Appendix A), with the questions divided into 3 groups. In the first group, employees had to answer a series of true or false questions; in the second group, multiple-choice questions were applied, where they selected the options that best suited them; and finally, in the third group, a table with different microorganisms was presented, and they had to mark with a cross those that were familiar to them. The questions addressed various topics related to food safety, such as the use of uniforms, foodborne illnesses and their symptoms, hygiene practices, cross-contamination, heat treatments, handling of raw and cooked meals, and temperatures in the danger zone.

### 2.5. Theoretical and On-the-Job Training

The theoretical training content was tailored based on insights gleaned from the checklist results, assessment of handlers’ knowledge, and the initial batch of microbiological findings. This instructional segment, lasting a maximum of 1 h and 30 min over 2 days, was scheduled to accommodate both staff and managerial presence. During these sessions, key observations compromising hygiene and food safety made during work hours were underscored. Topics encompassed various aspects, such as cleaning and disinfection protocols, temperature management, food microbiology, traceability, and proper food-handling procedures. Additionally, guidance on addressing the previously administered questionnaire was provided.

The on-the-job training initiative commenced upon staff arrival and extended until the conclusion of lunch service, spanning 3 to 4 days, as needed, with sessions averaging 6 h each. Throughout this duration, behaviours were monitored, and corrective measures were recommended. Instances of improper practices were identified, followed by demonstrations of correct and safe procedures. A supportive approach was adopted to elucidate why certain practices posed risks to food safety. Visual aids, such as videos and images, were employed to engage and educate staff on appropriate hygiene practices. Furthermore, adjustments regarding food storage practices were implemented as necessary.

Approximately 50 workers received the food safety training.

### 2.6. Microbial Analysis

#### Sample Collection

The microbiological analyses were conducted at two separate time points: before (BT) and 30 days after training (AT). The analyzed data corresponded to a total of 608 samples collected from equipment, surfaces, and utensils (ESU) (n = 416); the surface of food handlers’ hands (n = 128); and drains in the food preparation area (n = 64). A more detailed description of the collected samples is provided in Table 3.

The sample collection of hand surfaces and ESU swab samples was carried out using a sterile swab moistened in a 10 mL solution of tryptone salt (Himedia, Mumbai, India). The swab was rubbed over the surface to be sampled for 20 s, longitudinally covering the delimited area. Drains were analyzed using a sterile dry sponge sampling bag with a write-on strip and containing a dry, biocide-free cellulose sponge (VWR, Leuven, Belgium). The stipulated areas varied according to the type of surface and could be 10, 25, or 100 cm^2^, as described in ISO Standards 18593 [17], ISO 8199 [18], and NF EN ISO 6887 [19]. The microbiological determinations are represented in Table 4.

All samples were kept in a thermal box and transported to the laboratory for analysis within a maximum period of 1 h. In the laboratory, the samples were kept under refrigeration (2 ± 0.5 °C) and analyzed for the next 2–3 h. 

To quantify the bacterial populations total and pathogens, the tube samples were then homogenized for 60 s using a stomacher. The homogenates were then serially diluted, and 1 or 0.1 mL portions of the diluted suspensions were poured-plated by incorporation or surface-plated on non-selective and selective agar plates. To quantify different groups of bacteria, the following media and conditions were used: plate count agar (Himedia, Mumbai, India) at 30 °C for 72 h for the TMB (ISO 4833-1) [20]; Baird–Parker agar (VWR, Leuven, Belgium) at 37 °C for 48 h and confirmation was carried out at the end through the positive coagulase test with Rabbit Plasma Fibrinogen (Biolife, Monza, Italy) for coagulase-positive Staphylococcus (ISO 6888-1) [21]; Tryptone Bile Glucuronic agar (Liofilchem, Roseto degli Abruzzi, Italy) at 44 °C for 24 h for *E. coli* (ISO 16649-2) [22]; Violet Red Bile Glucose agar at (VWR, Leuven, Belgium) 37 °C for 24 h for *Enterobacteriaceae* (ISO 21528-2) [23]; Chloramphenicol Glucose (Biolife, Monza, Italy) agar at 25 °C for 5 days for molds and yeasts [24]; and Chromogenic medium agar (CHROMAgar, Saint-Denis, France) for detection, isolation, and enumeration of *L. monocytogenes* at 37 °C for 24–48 h (ISO 11290-2:2017) [25]. The counting of typical colonies was carried out, and results were expressed in colony-forming units per square centimeter (CFU/cm^2^).

For the drains, the detection for *L. monocytogenes* was performed using the sponge sample in 225 mL of Fraser I (Liofilchem, Italy) at 30 °C for 24 h, then Fraser broth (Liofilchem, Italy) at 37 °C for 24 h, and then spread on a chromogenic medium agar (ISO 11290:1-2017/AFNOR Validation CHR-21/1-12/01) [26].

### 2.7. Data Analysis

To assess the microbiological quality levels, all of them were classified according to the standards as described in Table 5, where all samples were classified into 2 levels: non-compliant and compliant.

The microbiological data obtained from the analyses carried out on the four restaurants were statistically analyzed using Statistica 12.0 software (StatSoft, Tulsa, OK, USA). Student’s statistical *t*-test was used for independent variables to compare the questionnaire values with the level of education of the employees. ANOVA and Kruskal–Wallis statistical tests were used to compare the evolution of microbiological results based on various parameters, such as the influence of training measures, sampling time, and surface type, and identify significant differences (*p* < 0.05). The ANOVA statistical test was used when the values followed a normal distribution, while the Kruskal–Wallis test was used for values with non-normal distributions.

## 3. Results and Discussion

### 3.1. Checklist

The results obtained from applying the checklist for each of the six included modules are presented in Table 6.

In module I (physical facilities and environment), the facilities construction, conception, and hygiene were evaluated. All four restaurants were found to have more conformities (58.7%) than non-conformities (29.9%). However, a lack of hygiene in the facilities was observed in three out of four restaurants. It should be noted that the ceilings were found to have mold and excess grease. This result was not in agreement with Ifeadike et al. [30] and Souza et al.’s [31] studies, where most of the analyzed restaurants were found to have good hygiene standards in their infrastructure, with 60% and 68%, respectively. 

In the four restaurants, there was no established handwashing procedure, and three of them lacked disinfectants, which led to cross-contamination between the poorly washed hands and the raw food being prepared. A similar observation was made in Bangladesh by Nizame et al. [32], where only 34% of establishments monitored had hand sanitizers. The non-conformity rate in this module (29.9%) was similar to the study by Da Cunha et al. [33], which reported an average rate of 34.1% (average between facilities and environmental hygiene).

In module II (food handlers), the food handlers’ clothing and hygiene habits were evaluated. Restaurants A, B, and C had a higher number of non-conformities (51.7%) compared with conformities (42.2%). The reason behind this was the incorrect use of gloves and the existence of skin and wounds on the manipulator’s hands.

No restaurants had continuous supervision to ensure correct handling and hygiene practices, and handwashing was not done with the required frequency or in the correct manner, negatively impacting the microbiological load shown further. The average non-conformity of this module was 51.7%, which is a result higher than that found in the study by Souza et al. [31], which obtained a non-conformity rate of 26.4% for the same module.

In Andrade et al.’s [34] study, restaurants with a non-conformity rate of 48.5% were classified as high risk for food safety. This suggests that the restaurants enrolled in this study would fall into a high-risk category related to food safety.

In module III (equipment and utensils), contact surfaces (countertops and cutting boards), equipment, and utensils were evaluated. It was observed that restaurants A, B, and D had surfaces in poor condition, with a lot of deep cuts facilitating the accumulation of pathogenic microorganisms, and all restaurants had poorly sanitized surfaces, which contributed to the contamination of the food being prepared. Utensils were not separated or identified by area, contributing to a lot of cross-contaminations, and in restaurants A and B, the utensils were made of unsuitable material, like wood, potentializing the accumulation of microorganisms inside the cracks. Regarding equipment, the main problem was a lack of hygiene in general. In all the restaurants, there was inadequate hand hygiene between handling raw and cooked foods, and employees did not frequently wash their hands during their work shifts. The average non-conformity rate for module III was 63.7%.

In module IV (receipt and storage), the non-conformities detected were focused on how foods were stored, with, for instance, potatoes in direct contact with the floor. Foods were stored without identification of the batch, which made it difficult to identify perishable foods in terms of the expiration date and their traceability. Additionally, temperatures in the cold storage were not controlled or recorded. The average non-conformity rate was 28.8%, which is a lower value than that found by Andrade et al. [34], who obtained a non-conformity rate of 48.1% for high-risk restaurants. Souza et al. [31], however, obtained a conformity rate of 77.3% for the same category of aspects analyzed, approaching the value obtained in this study, which was 65.6% conformity. Souza et al. [31] obtained a conformity rate of 77.3% for the same analyzed aspects category, which is a result closer to our study (65.6% conformity).

In module V (preparation, cooking, pantry, and distribution), all the practices involving preparing, handling, cooking, and food trajectories were evaluated. Non-conformities were mainly related to the lack of separation between the handling of raw and cooked foods, which was partially due to there being no separate routes for these two types of foods. The difficulty in having staff dedicated only to the preparation of raw foods was also observed, which, when combined with a lack of proper hand hygiene routine, could lead to cross-contamination of the food. The presence of chemicals used in cleaning and disinfection procedures in the food preparation sector was also an important detected non-conformity. Despite what was mentioned before, the average conformity rate in this module was 64.5%, which is very similar to the 67.6% conformity rate obtained by Da Cunha et al. [33].

In module VI (quality control), documentation and traceability were evaluated. Flaws were observed in all restaurants regarding signage and the collection of “safety samples” of meals prepared that day. Restaurants A, B, and C had problems with temperature records, traceability of foods, lack of secondary expiration dates (after first food handling), and the absence of microbiological control. The average classification for conformities in this module was 41.2%. Da Cunha et al. [33] obtained a conformity rate of 51.9% for the same assessed category of parameters (average values for documentation, traceability, and pest control). Higher conformity rates were found by Souza et al. [31] and Andrade et al. [34], with 69.0% and 62.9%, respectively, for this kind of evaluated parameter. 

#### Qualitative Classification

The restaurants were classified as “very good”, “good”, “acceptable”, and “not acceptable”, as described in Table 7 of the materials and methods. With this, in Table 5, you can find the ratings obtained by the different restaurants evaluated in this study.

Restaurant C received a rating of “not acceptable”, with a score of 46.4%, while restaurants A, B, and D obtained an “acceptable” rating with scores of 52.4%, 58.1%, and 66.2%, respectively. We consider the results obtained by us to be concerning and indicative of a lack of organizational concern and implementation of good hygiene and food safety practices.

### 3.2. Questionnaire

Table 8 represents the sociodemographic characteristics of the surveyed staff (n = 19), regardless of the restaurants. Out of the employees from the four restaurants, only 19 agreed to respond to the questionnaires, including 8 males (42.1%) and 11 females (57.9%). The ages of the respondents varied between 20 and 59 years. Regarding the job role, 12 (63.2%) employees were involved in food preparation and cooking, while 7 (36.8%) worked as waitstaff.

The questionnaire also revealed that 68.4% of the employees received training on food safety, which was significantly higher than the results obtained in the study by Chen et al. [35]. In their study of workers in China, Peru, and the United States, less than 20% of Chinese and Peruvian employees received food safety training. However, among food handlers in the United States analyzed in the study, the percentage of workers with food safety training was significantly higher, exceeding 90%. In another study conducted in Nigeria by Ifeadike et al. [30], 32.1% of food handlers did not have training in hygiene and food safety, which is a value very close to ours at 32.6%.

The results indicate that there was no significant difference between training and the percentage of correct answers to the questionnaire (*p* = 0.405). In other words, it was not possible to establish any relationship between the respondents’ knowledge level and their level of training.

Table 9 represents the percentage values of correct and incorrect answers regarding personal hygiene and behaviors.

The average value of correct answers obtained was 69.3%, and for incorrect answers, it was 30.7%. The respondents faced difficulty while answering the questions. It is incorrect to say that “burns and wounds can only be covered with band-aids” because gloves also need to be worn while applying the band-aids. A total of 14 (73.7%) of the respondents gave an incorrect answer. Stratev et al. [36] found that 97.8% of their respondents correctly answered that wounds should be covered with gloves.

On the other hand, almost all respondents were aware that hand hygiene is a crucial way to prevent food poisoning, and that maintaining a high level of hygiene is mandatory for all food handlers.

In Table 10, the frequencies of the answers given by the respondents to questions related to food poisoning are presented. These questions aimed to ascertain whether the respondents could correlate various symptoms with foodborne illnesses.

Out of the 19 respondents, 8 (42.1%) were able to identify vomiting, diarrhea, fever, and abdominal pain as symptoms of food poisoning. Among the remaining 11 (58.9%) of respondents, 7 (38.8%) recognized only vomiting and diarrhea, 1 (5.3%) recognized fever, and 5 (26.3%) recognized nausea and abdominal pain. In the study by Jianu et al. [37], 77% of food handlers were able to relate the mentioned symptoms as resulting from a foodborne illness.

Table 11 represents the frequency of correct and incorrect answers regarding cross-contamination, good food-handling practices, and foodborne disease agents. In this group of questions, an average value of 77.9% of correct answers were obtained and 22.1% of incorrect answers.

It seems important to mention that for the question “freezing food allows the elimination of pathogenic microorganisms”, only 11 (57.9%) of the respondents answered correctly. This result is consistent with previous studies by Ergönül et al. [38] and Gkana and Nychas [39]. They obtained correct response rates of 68% and 66.6%, respectively, for the same question. Also, in another study conducted in China by Gong et al. [40], only 12.4% of respondents knew that freezing does not kill bacteria in food.

In this group, there was also difficulty in responding to the question about whether it is possible to tell that a surface is contaminated just by observation. With a correct response rate of only 11 (57.9%). Consistent with the study by Gkana and Nychas [39], 78.9% of the respondents agreed that boards with many perforations should be replaced or sanded. In the study mentioned, a slightly higher percentage of 83.21% was found.

Finally, the knowledge of restaurant employees about pathogenic microorganisms of food origin was evaluated, and the results are presented in Table 12.

Among the surveyed respondents (n = 19), 18 (94.7%) were aware of the existence of *Salmonella* sp., and this microorganism was recognized as a pathogenic microorganism. The majority of respondents were unable to recognize the other presented microorganisms as pathogenic microorganisms or their relationship with foodborne illnesses. None of the respondents recognized *Yersinia* or *Campylobacter jejuni* as pathogenic microorganisms.

On average, there were 1.74 (out of 8) known microorganisms per person, which is almost 2 (out of 8) microorganisms per person. In the study conducted by Saeed et al. [41], 16.3% of respondents could associate *Staphylococcus aureus* with foodborne illnesses, which is a result close to the 10.5% in our study.

### 3.3. Microbiological Results

#### Equipment, Surfaces, and Utensils

The results related to the use of the ANOVA statistical analysis are presented to compare the effects of employee training in food hygiene and safety on the microbiological counts obtained for each of the restaurants under study.

Based on the analysis of Table 13, it was observed that for *Enterobacteriaceae*, restaurants A and C showed statistically significant differences (*p* ≤ 0.01) in the microbiological counts of equipment, surfaces, and utensils after training. For TMB, employee training significantly influenced the microbiological results in all restaurants (*p* ≤ 0.05). In the case of the mold counts, restaurants A and C also exhibited significant differences (*p* ≤ 0.05). Regarding the yeast counts, all restaurants showed significant differences (*p* ≤ 0.05).

The high mean values for microorganisms before the employee food safety training were a concern because they surpassed the food safety limits by a lot. In this study, the surfaces (cutting boards and raw meat preparation counters) were the most contaminated with the *Enterobacteriaceae* and TMB load throughout the study (Appendix A). This result was also obtained in the studies of Bukhari et al. [42] and Hartantyo et al. [43], where the cutting boards for raw meat preparation were the most contaminated. This high contamination can be simply explained by the fact that after the cutting of raw meat, the cutting boards and counters were left alone with blood throughout the working session without cleaning or only being cleaned with water. Another important point to refer to is that most of the cutting boards in all restaurants had deep cuts, which allowed for the accumulation of big loads of microorganisms. Some utensils, like cutting knives, were also a vehicle of microorganism’s contamination and were responsible for a lot of cross-contamination since they were used to cut almost everything without proper cleaning between tasks. After training, the staff started disinfecting the cutting boards and counters with the correct products at the correct time and some of the cutting boards that presented too many deep cuts or were worn out were replaced with new ones. The utensils also started being disinfected between different tasks with the correct product and were delimited as preparation, confection, and serving areas.

In previous studies, reductions in microbiological load were observed after employee food safety training. In a study reported by Levy et al. [44], a reduction of 22.6% in microbiological load was observed after training. A significant reduction in the microbiological counts was also observed after training in the Soares et al. [32] study.

Our study achieved reductions in microbiological count close to the minimum of 80%.

This significant decrease in microbiological levels can be attributed to high levels of microbiological contamination due to the lack of proper hygiene practices during work before the training. Even when some hygiene practices were put in place, they were often done incorrectly, failing to improve the food safety parameters.

### 3.4. Presence of Listeria monocytogenes in the Drains

During our investigation, LM was isolated from the drainage grates on the kitchen floors of restaurants, which was attributable to poor hygiene practices observed before the training period. According to Table 14, LM was detected before the training assessments as follows: three out of eight analyses for restaurant A, two out of eight for restaurant B, one out of eight for restaurant C, and two out of eight for restaurant D. In a study conducted by Toro et al. [45], LM was consistently found in the drains during all visits to the catering establishment.

Following the training period, none of the collected drain samples tested positive for LM, confirming the efficacy of the suggested measures, including the use of a specific disinfectant product with mechanical action. This successful removal of LM from the drains and prevention of its accumulation in these areas contrasts with findings in other studies. Britton et al. [46], for instance, assessed LM presence before and during the COVID-19 pandemic, with the latter characterized by stricter hygiene and disinfection protocols. Despite heightened cleaning, sanitization, and personal hygiene measures, contradictory results were obtained, indicating a persistent prevalence of LM in establishments. This underscores the necessity of implementing a targeted and comprehensive hygiene protocol specifically addressing LM contamination.

### 3.5. Analysis of Food Manipulators’ Hands

The results of the Kruskal–Wallis statistical test for the microbiological counts obtained from the analysis performed on the food manipulators’ hands can be found in Table 15.

In each of the four evaluated restaurants, significant differences in the microbiological load of food handlers’ hands before and after training were observed.

In restaurant A, significant reductions were observed in the average microbiological load of ENT, TMB, and molds following training. This indicates that the training had a significant impact on improving the hand hygiene of the food handlers. In restaurant B, significant differences were found in counts of ENT and TMB, with huge reductions in the average values after training. Impressive reductions were observed in the average counts for TMB and yeast in restaurant C, while significant differences were found in TMB and CoPS counts in restaurant D, with sharp reductions in the average values. It is worth noting that none of the food handlers tested positive for EC, which is a reassuring sign.

According to Table 15, the significant mean difference values indicate that there was a reduction in microbiological load of over 80%. This is an excellent outcome, which suggests that the theoretical and practical training was crucial in changing staff behavior. It was observed that the staff were not washing their hands correctly before the food safety training. They would only wash with water and, in some cases, forget to wash during service and between tasks, leading to cross-contamination between raw and cooked food. The low number of staff working in each of the four restaurants partially explains why it was almost impossible to have separated people for different tasks, which highlights the importance of proper hand washing to prevent cross-contamination. It must be emphasized that a lack of personal hygiene was also identified in Reynolds and Dolasinski et al.’s [47] study as the main cause of non-compliance with food safety standards.

Restaurants B and D had the highest presence of CoPS bacteria, which was expected, as the food handlers in these restaurants had open wounds on their hands. This made them more likely to be carriers of CoPS, which poses a significant risk when they handle food. After the staff’s training, restaurant D saw an 80% decrease in CoPS counts, while it was still present in restaurant B, which is indicative that the effect must be very dependent on the interest and will of the staff.

In general, after undergoing food safety training, the staff recognized the importance of using proper disinfectants for washing their hands was recognized. As a result, they started washing their hands more frequently at the correct times, which led to a reduction in the microbiological load referred to above. Also, they comprehended the importance of washing their hands at the beginning of work, when they arrive at the workplace, or when they leave the kitchen or manipulate the trash, which were things that were not considered before the food safety training. Insfran-Rivarola et al.’s [10] study concluded that the establishments that received theoretical and practical training combined, obtained more successful results than the ones that only had theoretical training. This can partially explain the good results (80% reduction) achieved in our study.

### 3.6. Effects of Training on Microbial Counts

The restaurant owners mentioned that their employees had previously received theoretical training. However, the study suggests that the employees who underwent exclusively theoretical training referred to this approach as tedious and not conducive to behavior change. Therefore, our on-the-job training proved essential for identifying and correcting incorrect practices during work shifts. Since the researcher was with the employees during working hours, they were able to identify the shortcomings in real-time and teach how to correct them, such as proper handwashing and disinfection, correct attitudes toward wound protection, storage of food in refrigerators and freezers, use of utensils categorizing between preparation, cooking and ready-to-eat food zones, measuring the temperature inside the food, and other behaviors that can prevent cross-contamination and unsanitary final products. By integrating theoretical and practical training in a coordinated manner, it was possible to promote an effective change in the habits and knowledge of the team, significantly contributing to improving food safety in the workplace. According to the statistical results demonstrated previously, the analyses indicate that there was a significant reduction in the microbial load across various areas following the implementation of the training. Therefore, the results presented stem from the combined effect of the training conducted rather than individualized efforts.

### 3.7. Limitations of Study

This study encountered several limitations that impacted its scope and applicability. One such limitation was the limited participation of establishments, which was attributed to a lack of recognition of the importance of food safety studies in driving improvements. Additionally, entrenched behaviors among the staff, particularly food handlers, were observed, with resistance to change evident even when acknowledging its necessity. To address these limitations, a practical “on-the-job” training method was implemented to integrate the researcher with the team, which aimed to enhance trust and collaboration. However, it is acknowledged that this approach may require sensitivity and adaptive strategies to ensure effective team engagement. While these limitations may influence the depth of the findings, awareness of them enables the development of strategies to mitigate their effects and improve the study’s quality.

## 4. Conclusions

Using checklists, our evaluation of the restaurants indicated a critical need for improvement in food safety practices and hygiene conditions, with compliance rates ranging from 52.4% to 66.2%. Although the majority of employees had received theoretical training in hygiene and food safety before this study, our analysis did not reveal significant differences between trained and untrained personnel, suggesting a possible gap between theoretical training and practical implementation. On-the-job training, along with the theoretical component in specific sequences, proved invaluable in correcting incorrect practices during work periods and providing contextual learning opportunities.

The microbiological results after the joint training showed significant reductions in microbial counts, highlighting the effectiveness of targeted interventions in mitigating microbiological risks.

This study underscored the critical need for robust infrastructure, self-control mechanisms, and mandatory employee training to maintain food safety standards and protect consumer health in restaurants.

## Figures and Tables

**Figure 1 microorganisms-12-00825-f001:**
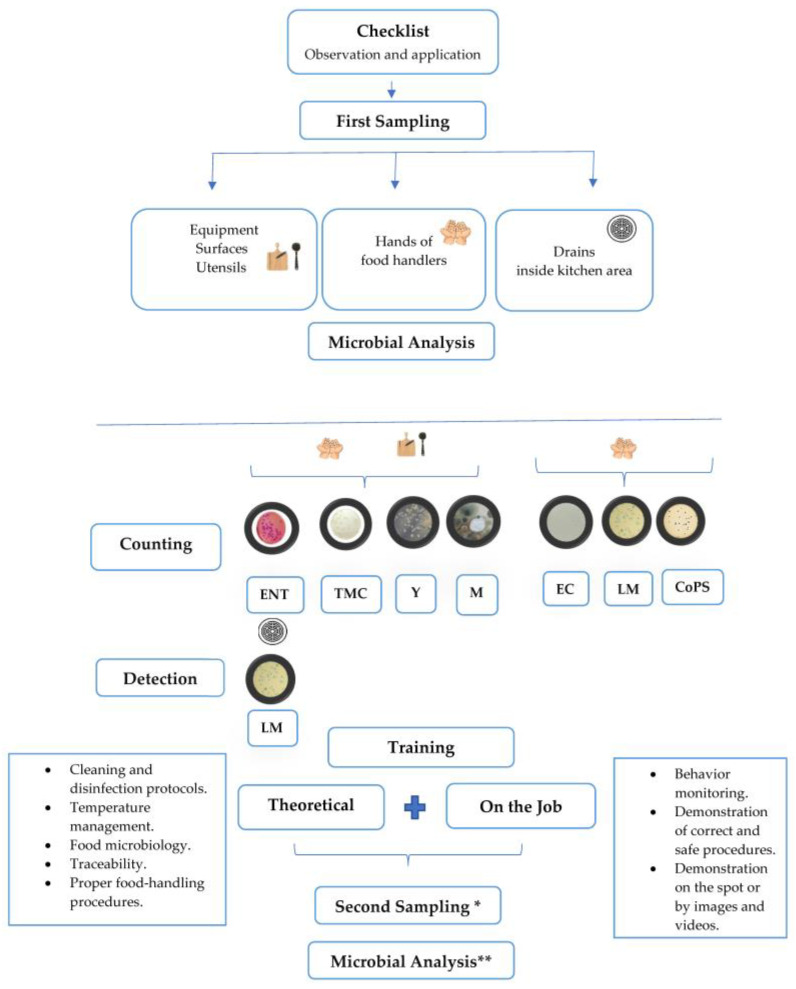
Experimental design conducted at restaurants in the study. ***** The second sampling was done in the same way as the first sampling. ****** The final microbiological analysis was effectuated the same as the first one. **ENT**—*Enterobacteriaceae*; **TMB**—total mesophilic bacteria; **Y**—yeast; **M**—mold; **EC**—*Escherichia coli*; **LM**—*Listeria monocytogenes*; **CoPS**—coagulase-positive *Staphylococcus*.

**Table 1 microorganisms-12-00825-t001:** Checklist module descriptions with the respective number of sub-items, number of items, module weight (W), and module constant (K).

Module	No. of Sub-Items	No. of Items	Module Weight (W)	Module Constant (K)
Physical facilities and environment	72	14	10	72
Food handlers	29	6	15	29
Equipment and utensils	20	4	15	20
Reception and storage	53	5	20	53
Preparation, cooking, pantry, and serving	69	12	20	69
Quality control	57	9	20	57

**Table 2 microorganisms-12-00825-t002:** Qualitative classification of the checklist applied to the four restaurants and the score intervals (%) based on Vieiros et al. (2007) [16].

Qualitative Classification	Score Intervals (%)
Very good	≥90%
Good	≥75% and <90%
Acceptable	≥50% and <75%
Non-acceptable	<50%

**Table 3 microorganisms-12-00825-t003:** Description and number (n) of samples before and after training for equipment (E), surfaces (S), utensils (U), food manipulators’ hands (H), and kitchen floor drains (D).

			Samples		
Training	E	S	U	H	D
	“Microwave, stove, oven”	“Countertop, cutting boards”	“Cooking knives, cutlery, tongs”	During work	Kitchen
Before	56	92	60	64	32
After	56	92	60	64	32
Total	112	184	120	128	64

**Table 4 microorganisms-12-00825-t004:** Microbiological determinations for detection and counting of each type of microorganism.

	Sampling	Detection	Counting
Indicators microorganisms			
ENT	ESU and hands		X
TMB	ESU and hands		X
MY	ESU and hands		X
Pathogenic microorganisms			
CoPS	Hands		X
EC	Hands		X
LM	Drain	X	X

ENT—*Enterobacteriaceae*; TMB—total mesophilic bacteria; MY—mold and yeast; CoPS—coagulase-positive *Staphylococcus*; EC—*Escherichia coli*; LM—*Listeria Monocytogenes*.

**Table 5 microorganisms-12-00825-t005:** Microbiological quality classification of meals served at restaurants. Adapted from M. Moragas et al. [27], K. Soares et al. [28], and Labović et al. [29].

Microorganisms	Non-Compliant	Compliant
ENT	>1 CFU/cm^2^	≤1 CFU/cm^2^
TMB	>100 CFU/cm^2^	≤100 CFU/cm^2^
EC	≥2 CFU/cm^2^	<2 CFU/cm^2^
MY	>10 CFU/cm^2^	≤10 CFU/cm^2^
CoPS	≥2 CFU/cm^2^	<2 CFU/cm^2^

ENT—*Enterobacteriaceae*; TMB—total mesophilic bacteria; MY—mold and yeast; CoPS—coagulase-positive *Staphylococcus*; EC—*Escherichia coli*; LM—*Listeria Monocytogenes*.

**Table 6 microorganisms-12-00825-t006:** Number (n) of conforming (C), non-conforming (NC), and not applicable (NA) for each checklist module, as well as the mean value.

Study Variables	Restaurant	C	NC	NA
		n (%)	n (%)	n (%)
Module I Physical facilities and environment	A	37 (51.4)	28 (38.8)	7 (9.7)
	B	52 (72.2)	13 (18.1)	7 (9.7)
	C	33 (45.8)	25 (34.7)	14 (19.4)
	D	47 (65.3)	20 (27.7)	4 (6.9)
	Mean (%)	58.7	29.9	11.5
Module II Food handlers	A	7 (24.1)	20 (69)	2 (6.9)
	B	12 (41.4)	15 (51.7)	2 (6.9)
	C	12 (41.4)	14 (48.3)	3 (10.3)
	D	18 (62.1)	11 (37.9)	0 (0)
	Mean (%)	42.2	51.7	6.0
Module III Equipment and utensils	A	7 (35)	13 (65)	0 (0)
	B	7 (35)	13 (65)	0 (0)
	C	7 (35)	13 (65)	0 (0)
	D	8 (35)	12 (60)	0 (0)
	Mean (%)	36.3	63.7	0
Module IV Receipt and storage	A	37 (69.8)	15 (28.3)	1 (1.9)
	B	38 (71.7)	14 (26.4)	1 (1.9)
	C	22 (41.5)	22 (41.5)	9 (17)
	D	42 (79.2)	10 (18.9)	1 (1.9)
	Mean (%)	65.6	28.8	5.7
Module V Preparation, cooking, pantry, and serving	A	46 (66.7)	15 (21.7)	8 (11.6)
	B	48 (70)	13 (18.8)	8 (11.6)
	C	36 (52.2)	20 (29)	13 (18.8)
	D	48 (69.6)	13 (18.8)	8 (11.6)
	Mean (%)	64.5	22.1	13.4
Module VI Quality control	A	23 (40.4)	33 (57.9)	1 (1.8)
	B	22 (38.6)	34 (59.6)	1 (1.8)
	C	16 (28.1)	40 (70.2)	1 (1.8)
	D	33 (57.9)	22 (38.6)	2 (3.5)
	Mean (%)	41.2	56.6	2.2

Legend: A, B, C, D—restaurant designations.

**Table 7 microorganisms-12-00825-t007:** Scores (%) and qualitative classifications of the restaurants A, B, C, and D.

Restaurant	Score (%)	Classification
A	52.4%	Acceptable
B	58.1%	Acceptable
C	46.4%	Not acceptable
D	66.2%	Acceptable

Very good (≥90%); good (≥75% and <90%); acceptable (≥50% and <75%); not acceptable (<50%).

**Table 8 microorganisms-12-00825-t008:** Sociodemographic characteristics of the staff that responded to the questionnaire (n = 19) by gender, age, working function, and theoretical training.

Characterization		n (%)
Gender	Female	11 (57.9)
	Male	8 (42.1)
Age	20–30	6 (31.6)
	31–40	3 (15.8)
	41–50	4 (21.1)
	≥50	5 (26.3)
Working function	Preparation and confection	12 (63.2)
	Waiter/waitress	7 (36.8)
Theoretical training	Yes	13 (68.4)
	No	6 (31.6)

**Table 9 microorganisms-12-00825-t009:** Frequency (%) of the number (n) of correct (%) and incorrect (%) responses about personal hygiene and behavior (n = 19).

Question	Correct n (%)	Incorrect n (%)
Hand hygiene is a way of preventing food intoxication	16 (84.2)	3 (15.6)
Burns and wounds can only be covered with band-aids	5 (26.3)	14 (73.7)
Only food manipulators may clean their hands after coughing	7 (36.8)	12 (63.2)
All workers must maintain a high level of hygiene	19 (100)	0 (0)
The uniform can only be dressed inside the working room	13 (68.4)	6 (31.6)
Handwashing should be done before cooking, whenever tasks are changed, at the end of cooking, and whenever needed	19 (100)	0 (0)
Mean	69.3	30.7

**Table 10 microorganisms-12-00825-t010:** Frequency (%) of the number (n) of responses about food intoxication and its symptoms.

Study Variables		n (%)
“Symptoms associated with food illness”	Vomiting and diarrhea	7 (38.8)
	Fever	1 (5.3)
	Nausea and abdominal pain	5 (26.3)
	Muscle aches	0 (0)
	All the above	8 (42.1)
“The appearing of the symptoms from a food poisoning may take __”	Minutes	2 (10.5)
	Hours	8 (42.1)
	Days	1 (5.2)
	All the above	5 (26.3)

**Table 11 microorganisms-12-00825-t011:** Frequency (%) of the number (n) of incorrect (%) and correct (%) responses about cross-contamination, good food manipulation practices, and agents.

Question	Correct n (%)	Incorrect n (%)
The temperature danger zone for the growth of microorganisms is between 5 and 65 °C	14 (73.7)	5 (26.3)
*Salmonella* is present only in eggs	14 (73.7)	5 (26.3)
Freezing food eliminates pathogenic microorganisms	11 (57.9)	8 (42.1)
Cooked foods cannot be responsible for food poisoning	14 (73.7)	5 (26.3)
Bags of carrots and potatoes can be placed directly on the floor	17 (89.5)	2 (10.5)
Raw fish/meat can be left at room temperature on the counter for more than 30 min	17 (89.5)	2 (10.5)
It is possible to tell if a surface or food is contaminated just by observation	11 (57.9)	8 (42.1)
Cooked and raw food should be handled with different utensils and stored in different places	18 (94.7)	1 (5.3)
The use of colour-coded cutting boards helps differentiate the types of foods that can be prepared on them, preventing cross-contamination	17 (89.5)	2 (10.5)
Cutting boards should be sanded or replaced when they have many perforations or cuts	15 (78.9)	4 (21.1)
Chemicals can be stored next to raw materials	15 (78.9)	4 (21.1)
Mean (%)	77.9	22.1

**Table 12 microorganisms-12-00825-t012:** Frequency (%) of the number (n) of “yes” and “no” responses about knowledge of pathogen microorganisms.

Microorganism	Yes	No
	n (%)	n (%)
*Salmonella*	18 (94.7)	1 (3.3)
*Clostridium botulinum*	3 (15.8)	16 (84.2)
*E. coli*	4 (21.1)	15 (78.9)
*Yersinia*	0 (0)	19 (100)
*Campylobacter jejuni*	0 (0)	19 (100)
*L. monocytogenes*	3 (15.8)	16 (84.2)
*Bacillus cereus*	1 (5.3)	18 (94.7)
*Staphylococcus aureus*	2 (10.5)	17 (89.5)

**Table 13 microorganisms-12-00825-t013:** Mean values (CFU/cm^2^) and standard deviation (Sd) of the ENT, TMB, M, and Y counts in the 4 restaurants (R) before and after training (BT and AT) of the staff and comparative analysis with ANOVA statistic test (*p*) of the effectiveness of the training (Tr) in each restaurant with the percentage of microbiological load reduction (mean dif.).

R	Tr	ENT		TMB		M		Y	
		Mean (Sd)	*p* (Mean dif.)	Mean (Sd)	*p* (Mean dif.)	Mean (Sd)	*p* (Mean dif.)	Mean (Sd)	*p* (Mean dif.)
A	BT	8.78 (25.7)	0.0006 (−98%)	418.69 (1377.0)	0.00001 (−99%)	88.38 (402.2)	0.0080 (−99%)	52.62 (173.4)	0.00001 (−100%)
	AT	0.17 (0.9)		5.44 (25.5)		0.96 (2.8)		0 (0)	
B	BT	216.84 (1414.4)	0.06 -	2051.58 (6119.3)	0.0001 (−99%)	43.96 (155.8)	0.78 -	253.55 (1086.0)	0.0318 (−97%)
	AT	0.23 (1.0)		10.82 (44.0)		1.73 (3.3)		6.90 (26.0)	
C	BT	42.84 (202.8)	0.0094 (−99%)	1913.31 (9008.8)	0.00001 (−97%)	16.79 (70.0)	0.0256 (−99%)	365.76 (1009.3)	0.0001 (−89%)
	AT	0.14 (0.4)		63.38 (340.9)		0.15 (0.8)		39.60 (178.1)	
D	BT	0.72 (3.5)	0.77 -	136.67 (685.5)	0.00001 (−80%)	0.21 (1.0)	0.72 -	30.17 (152.1)	0.0006 (−97%)
	AT	3.07 (14.4)		27.48 (166.0)		0.15 (0.6)		0.88 (6.2)	

Legend: non-significant (*p* > 0.05); significant (*p* ≤ 0.05); very significant (*p* ≤ 0.01); ENT—*Enterobacteriaceae*; TMB—total mesophilic bacteria; MY—mold and yeast.

**Table 14 microorganisms-12-00825-t014:** Presence (+) or absence (−) of *Listeria monocytogenes* on the drains for each restaurant in the harvest before (*n* = 8) and after employee training (*n* = 8).

	Samples	
Restaurant	BT	AT
A	+ ***	−
B	+ **	−
C	+ *	−
D	+ **	−

Legend: *—1 positive sample; **—2 positive samples; ***—3 positive samples.

**Table 15 microorganisms-12-00825-t015:** Mean values (CFU/cm^2^) and standard deviation (Sd) of the ENT, TMB, MY, and CoPS in the 4 restaurants (R) before (1) and after (2) employee training (Tr), and a comparative analysis with the Kruskal–Wallis statistic test (*p*) of the effect of the training in each restaurant (R) and the percentage of microbiological load reduction (mean dif.).

R	Tr	ENT		TMB		M		Y		CoPS	
		Mean (Sd)	*p* (Mean dif.)	Mean (Sd)	*p* (Mean dif.)	Mean (Sd)	*p* (Mean dif.)	Mean (Sd)	*p* (Mean dif.)	Mean (Sd)	*p* (Mean dif.)
A	1	13.1 (20.8)	0.0023 (−99%)	1381.5 (2489.5)	0.00001 (−99%)	148.8 (307.1)	0.006 (−99%)	327.7 (592.8)	0.16 -	1.3 (3.5)	0.54 -
	2	0.03 (0.1)		16.6 (17.08)		0.3 (1.0)		24.1 (33.9)		3.6 (14.5)	
B	1	2.4 (5.0)	0.0028 (−100%)	553.0 (776.8)	0.0339 (−36%)	4.3 (9.9)	0.37 -	12.0 (29.4)	0.37 -	52.8 (191.7)	0.37 -
	2	0.00 (0)		355.0 (649.7)		0.5 (1.4)		38.0 (103.8)		83.8 (194.0)	
C	1	49.5 (151.5)	0.20 -	1495.7 (2507.6)	0.0014 (−95%)	24.0 (61.6)	0.07 -	2416.1 (5102.5)	0.0004 (−99%)	0.0 (0.0)	0.31 -
	2	0.1 (0.4)		80.5 (237.2)		0.0 (0.0)		1.5 (4.4)		5.5 (22.0)	
D	1	0.2 (0.6)	0.20 -	1289.7 (1562.3)	0.0339 (−92%)	0.3 (1.0)	0.14 -	0.8 (152.1)	0.37 -	281.1 (478.9)	0.014 (−80%)
	2	0.5 (1.0)		99.9 (140.6)		1.0 (1.8)		1.5 (2.9)		55.0 (220.0)	

Legend: non-significant (*p* > 0.05); significant (*p* ≤ 0.05); very significant (*p* ≤ 0.01); ENT—*Enterobacteriaceae*; TMB—total mesophilic bacteria; M—mold; Y—yeast; CoPS—coagulase-positive *Staphylococcus*; EC—*Escherichia coli*; AT—after training; BT—before training.

## Data Availability

Data are contained within the article and Appendix A.

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
