# Peer review of "Evaluation of the Effects of Food Safety Training on the Microbiological Load Present in Equipment, Surfaces, Utensils, and Food Manipulator’s Hands in Restaurants"

_microorganisms, 2024, doi:10.3390/microorganisms12040825_

Round 1

Reviewer 1 Report

Comments and Suggestions for Authors

The research entitled "Evaluation of the Effects of Food Safety Training on the Microbiological Load Present in Equipment, Surfaces, Utensils, and Food Manipulator’s Hands in Restaurants" aimed to verify the hygienic conditions in restaurants before and after training (theoretical and practical) and to evaluate the effect on microorganism counts in equipment, surfaces, and utensils. The topic is crucial as a significant portion of foodborne disease outbreaks occur due to these shortcomings in food service establishments, necessitating control measures. Knowledge in this area contributes to a better understanding of applicable prophylactic measures for preventing foodborne disease outbreaks. However, the manuscript is not well-written and prepared.

The English language usage MUST be improved. Clearly, the manuscript was written in a language (Portuguese?) and later translated by the authors, resulting in slightly confusing writing not entirely conforming to English language norms. I suggest revision by a native English speaker. For instance, terms like "written visit" in line 77 and "mother suspensions" in line 158 require attention.

There is no mention of approval by a human ethics committee anywhere in the manuscript. Was the research approved by a committee? Include this information in the appropriate section.

The abstract section requires improvement. The authors provide a lengthy introduction and preliminary results before training but omit a significant portion of the main research findings directly related to the primary objective. The results of the training's effect should be clearly and precisely presented. Additionally, a conclusion should be included in the abstract.

The introduction is extremely brief and does not provide the necessary background for understanding the study. While the authors mention HACCP, it is not directly related to the research conducted. They fail to present information on contamination in these types of establishments, data on outbreaks in such establishments, the indicator microorganisms evaluated in the samples, and the legislation mandating GMP adoption in these establishments. This section requires a complete overhaul to be engaging to international readers. Additionally, the paragraph between lines 62 to 65 should be removed, as research results should not be anticipated; instead, the introduction should conclude with the research objective.

In the Materials and Methods section, the authors should include a descriptive illustration of the experimental design to facilitate understanding of the study, which is not clear throughout this section. They should also include all checklists and questionnaires used as supplementary files, as they may be useful for other research in the area.

Regarding sample collection for microbiological analyses (paragraph in line 134), the authors should provide a more detailed description of the samples collected. Providing another supplementary file may help readers understand better.

Additionally, the origin location of the manufacturer for all inputs used should be included.

In Table 3, do the authors refer to "detection" when using the term "Search"?

The authors should include references for all microbiological protocols used, not just cite the ISO standard number used, for example. Reference the entire methodology.

The authors claim to have detected the species S. aureus, but according to the described methodology, they only detected coagulase-positive Staphylococcus. Review this mistake throughout the manuscript. Not all coagulase-positive Staphylococcus are S. aureus.

The theoretical training should be more thoroughly described. How many times was it conducted? How long did it last? What methodologies were used? Describe as comprehensively as possible. The same applies to practical training.

Regarding statistical analyses, I would like to see information on which modified practices after training most influenced the reduction in microorganism population counts. The authors have ample data and can perform this analysis to show which modified practices or generated knowledge had the most influence on improving overall hygiene in the establishments.

In the Results and Discussion section (topic 3.1 and others), the authors only compare results with other studies but do not discuss the inherent risks of lacking hygienic practices. Information about the consequences of lacking these practices and possible improvements should be included. The discussion is merely comparative and poorly developed.

Throughout this section, numerical values should be included in addition to percentage values. For example, "12 (X%) respondents claimed to know...".

Additionally, the presentation of checklist and questionnaire results is extremely lengthy and tiresome, requiring a restructuring to become engaging.

In the results section, the authors claim that certain equipment was more contaminated, for example, but no data is presented in this section, and it was not mentioned in the Materials and Methods section.

This section (topic 3.3.) should be the main focus but is weakly explored by the authors. Topic 3.4 seems unnecessary as it was not mentioned previously and appears abruptly and suddenly in the manuscript. It should be merged with the previous topic or have its title modified.

The authors should also discuss the differences between the effects of theoretical and practical training. Although both forms were conducted, no data is presented and discussed regarding the differences between these methodologies. A paragraph containing the study's limitations is also necessary.

Overall, the tables require better formatting and also detailing of the units of each column/row to facilitate result visualization.

The conclusion should be completely reformulated as it consists of a repetition of results. Only succinctly conclude what the results actually allow concluding, using a maximum of two paragraphs.

Minor remarks

Line 16 – Verify the use of the term "equip".

Line 34 – The use of references is not in line with journal standards. It should be: "[1,2]". Double-check this usage throughout the manuscript.

Line 37 – "In 2022," Line 46 – Use the abbreviation (HACCP) as the term has already been introduced.

Line 54 – Remove the reference using the authors' names.

 Lines 80, 81 – Remove "us". In Table 1, clarify what "module constant" means. Also, improve the title to be self-explanatory.

Line 152 – Repeated paragraph.

Line 155 – Follow journal norms for reference usage.

Line 162 – CFU – Standardize throughout the manuscript using uppercase letters.

Lines 300-301 – Please write scientifically.

Line 352 – "Food intoxication" is not synonymous with "foodborne illness".

Lines 405-407 – Review sentence construction. Write scientifically.

Table 2 requires restructuring for better understanding (N=8X2 ????)".

Comments on the Quality of English Language

Require extensive revision.

Reviewer 2 Report

Comments and Suggestions for Authors

Training (theoretical and on the job) of the staff of 4 restaurants was evaluated by comparing the microbiological loads of restaurant surfaces, drains and the hands of food handlers before training and 30 days after training.

General

Line 197 The word "formation" should be replaced with "training" throughout the manuscript.

Lines 313 and 314 What is the total number of staff in the 4 restaurants? How many staff were trained by the researchers?

Specific

 Line 23 Delete "positive" after Staphylococcus aureus.

Lines 34 and 35 Foodborne illnesses often occur due to improper handling, preparation storage, transportation, and sanitation in restaurants.

Lines 58 to 60 Replace "perform microbiological analyses on" with "determine the microbiological loads of". Delete "order to assess the microbiological state of".

Lines 62 to 65 Delete. 

Line 104 Where is "K" derived from?

Line 136 Replace "The analysis was done using swabs in" with "The".

Line 137 Add "were swabbed" after "hands". Delete "always".

Line 252 Replace "their" with "there".

Line 441 Place "was isolated" after "Listeria monocytogenes"

Lines 451 to 461 Delete the repeated text

Comments on the Quality of English Language

The English in the manuscript can be improved. Some suggestions are made in the Comments section above.

Round 2

Reviewer 1 Report

Comments and Suggestions for Authors

The quality of the manuscript has been significantly improved. However, some considerations were not fully addressed.

I have an extreme concern regarding a certain aspect of this manuscript. Most concerning of all is that the authors conducted the research without prior approval from an ethics committee, and are now attempting to seek approval after the research has been conducted and the manuscript submitted to this journal. The editorial board of Microorganisms should determine how to proceed in this case.

Outstanding Corrections:

The authors failed to include study limitations at the end of the discussion.

There was no differentiation between theoretical and practical training aspects throughout the results, and it was not evaluated how each of them individually affected the microbiological counts. Conducting statistical analyses could be helpful for this differentiation, but apparently the authors do not have the means to perform them. However, including a discussion on this topic is essential in the article.

Appendixes must be converted as Supplementary Files in this article.

The conclusion still consists of a repetition of the results. This section requires a reformulation to become objective and applicable.

Line 89-90 – Staphylococcus coagulase-positive, to remain consistent with the conducted research.

Do not italicize mycotoxins. They are not etiological agents.

The legend of Figure 1 needs improvement to become self-explanatory. The same applies to almost all tables and figures in the manuscript. Provide complete descriptions.

Table 1 is inserted into the text without prior mention.

Replace "Search" with "detection" in the tables.

Line 404 – "you can see" – avoid this construction throughout the entire article. The same applies to the use of expressions containing "us" or "we."

Avoid using "p-value." Use only p = 0.001, for example. P-values should be presented and not simply stated as "<0.05." Review this throughout the manuscript.

Comments on the Quality of English Language

Minor editing.
